# Analysis of the Possibilities of Tire-Defect Inspection Based on Unsupervised Learning and Deep Learning

**DOI:** 10.3390/s21217073

**Published:** 2021-10-25

**Authors:** Ivan Kuric, Jaromír Klarák, Milan Sága, Miroslav Císar, Adrián Hajdučík, Dariusz Wiecek

**Affiliations:** 1Department of Automation and Production Systems, Faculty of Mechanical Engineering, University of Zilina, 010 26 Zilina, Slovakia; ivan.kuric@fstroj.uniza.sk (I.K.); milan.saga@fstroj.uniza.sk (M.S.); miroslav.cisar@fstroj.uniza.sk (M.C.); adrian.hajducik@fstroj.uniza.sk (A.H.); 2Faculty of Mechanical Engineering and Computer Science, ATH–University of Bielsko Biala, 43-309 Bielsko-Biała, Poland; wiecekd@ath.bielsko.pl

**Keywords:** tire inspection, deep learning, unsupervised learning, polynomial regression, laser sensor, defect detection, polar transform

## Abstract

At present, inspection systems process visual data captured by cameras, with deep learning approaches applied to detect defects. Defect detection results usually have an accuracy higher than 94%. Real-life applications, however, are not very common. In this paper, we describe the development of a tire inspection system for the tire industry. We provide methods for processing tire sidewall data obtained from a camera and a laser sensor. The captured data comprise visual and geometric data characterizing the tire surface, providing a real representation of the captured tire sidewall. We use an unfolding process, that is, a polar transform, to further process the camera-obtained data. The principles and automation of the designed polar transform, based on polynomial regression (i.e., supervised learning), are presented. Based on the data from the laser sensor, the detection of abnormalities is performed using an unsupervised clustering method, followed by the classification of defects using the VGG-16 neural network. The inspection system aims to detect trained and untrained abnormalities, namely defects, as opposed to using only supervised learning methods.

## 1. Introduction

In the mass production of tires, which is characterized by a large number of manufactured items, it is very difficult to conduct the final quality inspection, considering the visual and qualitative aspects of the tires before they are placed on the market. Qualitative aspects focus on materials, geometry, tire appearance, and final function, while the visual aspects of tires are mainly formal (but necessary) features, such as annotations, barcodes, or other features necessary to identify the product. Visual content without defects is representative of a high-quality product. To ensure high quality in the mass production of tires, it is necessary to obtain data from the manufacturing process and to digitize the final quality inspection before the product leaves the factory. The current trend is to apply processes from Industry 4.0, focused on automation, machine learning, sensory systems, digitization within manufacturing processes, data visualization, etc. [1]. Most applications of machine learning are based on integrating supervised methods, such as convolutional neural networks (CNN), deep learning, and regression, with unsupervised methods, such as clustering algorithms. A combination of final tire inspection and monitoring of the technological processes during production, makes it possible to capture any damaged product by classifying its defects and can even identify the possible origins of the defect. This paper is focused on capturing defects during the final inspection of tires. The final inspection of tires is defined based on the complex manufacturing processes in the rubber industry. The tire is considered the object of interest, which is compounded from various materials and has a rugged surface including letters, symbols, and so on, which makes it suitable for evaluating the possibility of using an inspection system. From the view of quality assurance, there are many requirements, due to the high demands during vehicle operations and safety of the vehicle operator and crew. Inspection tasks and requirements have mainly been described in terms of processing 3D data obtained from laser sensors into visual content using pattern recognition [2,3]. A closer description of tire defects has been described in a Ph.D. thesis [4] conducted by the Department of Automation and Production Systems at the University of Zilina. There are defined statements that categorize tire defects in terms of their origin and during the manufacturing processes. A catalog describing possible tire defects is generally available to employees at the inspection stand in the factory. To simplify categorizing the most frequent defects, we defined six basic categories of defects (Figure 1):Impurity with the same material as the tire material (CH01);Impurity with a different material from the tire material (CH02);Material damaged by temperature and pressure (CH03);Crack (CH04);Mechanical damage to integrity (CH05); andEtched material (CH06).

The defects can be captured by a camera, such as those manufactured by the Balluff company. These images, with a resolution of 12 Mpx, were integrated into the experimental inspection stand described in [2] and used in the paper “Design of laser scanners data processing and their use in a visual inspection system,” which will be presented at the ICIE2020 conference (2021). In this thesis, we demonstrate the capturing of defined defects through the use of MATLAB 2020a software (MathWorks company), for which a convolutional neural network (CNN) was trained using a transfer learning method based on the AlexNet network [5]. The achieved defect detection accuracy was in the range from 85.15% (CH04) to 99.34% (CH03).

Similar work regarding an inspection system for the tire industry has been described in [6], where the inspection system was constructed using image processing techniques, and the principle of processing data was described as obtaining 3D images, converting them into 2D images, and conducting analysis through two different approaches. The first approach combined discrete Fourier transform and K-means clustering, while the second approach utilized artificial neural networks. Discrete Fourier transform was used to detect the defects in 3D images by inferring patterns with spatial domains. K-means clustering separated the pixels into clusters, where complications occurred when defining the specific number of clusters to obtain relevant data. The second approach, in particular, used LSTM neural networks. It is possible to use a CNN [7]; however, due to the necessity of defining a large learning data set, it is difficult to perform this task. Therefore, it is impractical to adapt a CNN to every item occurring in a wide range of products. LSTM neural networks work in real-time on data from the inspected object. In the commercial field, the Tekna Automazione e Controllo company specializes in the detection of defects that occur on the tire sidewall; however, their publicly available information does not allow for the identification of the algorithms used to perform the inspection. In general, visual inspection of the surface is mainly based on the use of 3D-captured surfaces by a triangulation principle, where laser sensors are used as final devices; for example, the Keyence laser sensor–laser profilometer mentioned in [8,9,10], or a camera vision system supported by a laser beam [11,12,13].

Techniques using the visual data obtained from cameras and devices with laser beams are currently widely used [14], including those utilizing CNNs (supervised learning) or clustering methods (unsupervised learning). As mentioned above [15], the use of convolutional neural networks can be complicated, especially when there is a wide range of items to search for on the captured products. However, in the case of specific mass monotonous productions, they can be profitably designed: simply create the data set, then train the neural network to perform product inspection. Furthermore, there can be many complications when detecting defects. In [16], it was mentioned that the change in color scale from color to grayscale due to defects includes many different colors, which can be exploited to perform defect detection tasks in a shorter time. In [17], the authors described the use of a CNN for wafer defect detection, describing it as a feasible alternative to manual inspection, but there were still misclassifications. Another similar work is [18], where three models were used for wafer defect detection. The lowest detection accuracy was 94.63% for WDD-Net when detecting mechanical damage, while the highest accuracy was 100%, which occurred in many of the cases mentioned in the paper. In the category of mechanical damage, it is necessary to manage significant amounts of data. The other point relies on manually labeling defects to prepare training data for a supervised model. In [19], the use of a CNN for aircraft maintenance by defect detection was described. The authors described complications when collecting a data set for each defect that occurs in aircraft maintenance, to train the model. Other papers have generally described using a CNN or R-CNN to classify defects with accuracy higher than 90% [20,21,22,23,24].

In summary, CNNs show potential for defect detection. The associated results are good, but these systems were mainly in experimental use, not in practical deployment. Based on experiments, the use of such systems is feasible; however, improvement or re-designing the logic of inspection systems based on CNN methods is necessary. This is mainly due to the nature of supervised learning, as mentioned in [18], where a large amount of data must be captured and labeled, which is a difficult task, to detect the trained defects or those which are very similar. Defects in real-life conditions are not simple to categorize simply with a certain number of shapes, as the differences in defects are additionally manifested in terms of their position, lighting condition, and so on. The described systems work properly when defects in the analyzed data appear very similar to those in the training data set. In the case of different defects, there is a very low probability of detection. One study [25] has described methods for defect detection in steel products using various technologies applied to visual data captured by a CCD camera. The results obtained through using more methods led to the statement: “the fusion of multiple technologies is an expected trend,” which can be interpreted as a necessity to develop systems that combine more technologies. As such, prospective defect detection methods should be able to conduct unsupervised learning within the inspection system. For instance, defect detection on wafers using the K-means method has been described [26]. Non-industrial areas describing methods combining supervised and unsupervised approaches include threat detection workflow [27]. In the medical area, research has been published regarding the use of unsupervised methods for gathering biological data to perform tumor detection with MRI data [28] or clustering and testing based on IRIS [29].

A possible solution is developing a comparative system, where the recognition of defects would be replaced by a comparison of the inspected data with standard reference data without defects. This involves conditioning through use in controlled conditions, where strictly defined attributes of the inspected object are considered. This condition is fulfilled in industrial conditions, specifically in mass production. In this case, every product should be identical. In this manner, abnormalities can be found, being characterizable as objects of interest without information regarding what it means. Then, in the following step, CNN-based methods can be used to classify the abnormalities as defects or as acceptable elements. This paper focuses on describing a comparative method and data fusion to improve inspection performance, where geometrical data are necessary to identify the geometry of the captured object and to determine geometric abnormalities, while the camera system is used to obtain the visual characteristics of the captured surface. At present, systems that are able to recognize pre-trained patterns or letters in the visual content of data are available; for example, the OCR function (MATLAB), or specific functions in various Python libraries. For this purpose, “normalized” visual content is necessary. The term “normalized” involves the managed modification of data in order to orient it similarly to standard text in an affine manner. The standard procedure involves unfolding the captured rotary object using a polar transform—a process that may be automated. The objects captured by the camera may decrease in quality when using an inappropriate hardware setup, which can affect the accuracy of the inspection system. Based on these considerations and previous works, we defined the following hypotheses:

**Hypothesis** **1** **(H1).**
*Is there the possibility to automate the polar transform procedure of the captured part of the tire sidewall?*


**Hypothesis** **2** **(H2).**
*Is it possible to compensate for inaccuracies in visual data captured by camera due to the use of inappropriately set up hardware?*


**Hypothesis** **3** **(H3).**
*Are deep learning techniques applicable to defect detection in visual data from laser sensors?*


**Hypothesis** **4** **(H4).**
*What is an appropriate design and application for a hybrid system integrating unsupervised learning for defect detection on tire sidewalls?*


## 2. Materials and Methods

According to previous work [3,30,31], which has described methods for processing point clouds captured by laser sensors from the Micro-Epsilon company, multiple data can be modified and adapted, fusing them into one whole with the focus of obtaining better results in a defect inspection system. Previous work has mainly been oriented to processing data from laser sensors. Methods to transform geometrical data to 2D data, such as visual content, have been described. Such transformations are feasible in many ways. In this paper, the method described in [31] is considered.

### 2.1. Camera Vision

Geometric data are insufficient when mapping the visual aspects of scanned objects. Instead, conventional data obtained by cameras are considered optimal. In the market, standard cameras with a resolution from 1–12 Mpx (monochrome) or 1–5 Mpx (color) are available. For our purposes, 12 Mpx monochrome industrial cameras BVS CA-M4112Z00-35-000 from the Balluff company were used. The first test utilized pictures of the whole sidewall of the tire, as shown in Figure 2. When adapting to pictures obtained from laser sensors, it is necessary to perform a polar transform of the sidewall and unfold it from an annulus shape to a rectangular one. In the Python language, there exist a few suitable frameworks to perform this task. The processing consists of two operations: The first is the detection of the center of the sidewall, and the second is the unfolding. Center detection was performed using circle Hough transform (CHT), implemented in the OpenCV library as an optimized algorithm [32,33], see Figure 2 (right). The outputs were circles characterizing the borders of the sidewall and, thus, defining the annulus of the sidewall. The second step is unfolding the annulus using the polarTransform library. The function is performed using the four parameters defined in the previous step of circle detection. An example of the unfolding result is displayed in Figure 3. After analysis of Figure 3, there is a sinusoidal character to the picture. The reason for this is the eccentricity of the detected circle characterizing the tire, as shown in Figure 2. The fusion of linear pictures, that is, from the laser sensor and the unfolded tire sidewall, is possible, but there is no added value; in fact, fusion could decrease the detail and quality of the final picture. The solutions rely on a centered picture before the polar transform using parameters of the center base annulus. The second attribute is resolution in the columns of the unfolded picture. The size of the unfolded picture is 8224 × 791 pixels. The pixel count in the column is 791, in comparison to that of the laser sensor, with a maximum of 640 pixels in the column, as the laser sensor captured a smaller area. Therefore, when using a laser sensor scanning 1280 points, the unfolded picture is insufficient. The solution relies on capturing a smaller area of the tire or capturing the whole sidewall of the tire using a camera with a higher resolution.

To increase the resolution, we captured smaller areas of the sidewall, as displayed in Figure 4, with a size of 4112 × 3008 pixels. This method is suitable for capturing the sidewall during rotation during the process of scanning. In the next step, we performed the unfolding, the result of which is fused with the associated pictures from the laser sensor. Unfolding can be carried out by any feasible conventional method, such as those in the OpenCV library or the polarTransform library. This process can offer sufficient results, but there is no calibration process. To improve the quality of visual data, an appropriate system design to modify data should be conducted when setting up the capture hardware. In the experimental stand, attributes such as the position of the camera, lights, and so on were not rigorously managed. This resulted in the observed eccentricity and the angle having a non-linear radius. The angle ε indicates the angle created by the noncollinearity between the normal to the sidewall and the normal to the camera, resulting in a difference between the radii of the out-circle and in-circle of the annulus, as shown in Figure 5. In addition, eccentricity indicates the picture is not aligned with the axis of the sidewall. According to the above, it is necessary to define parameters to compensate for such attributes (i.e., angle, eccentricity, and axis centering of the picture), as shown in Figure 4 and Figure 5.

Preparation for capturing the sidewall area in the picture to unfold from involves defining the border edges, as displayed in Figure 4, where the purple pixels indicate the outer radius and the red pixels indicate the inner radius, while the green pixels characterize the values halfway between the red and purple pixels. According to the red and purple pixels, it is possible to define the parameters for unfolding.

The first step is to define the principle of the unfolding process, where pixels are chosen in a specific way and transformed to a certain row (red) or column (blue), as illustrated in Figure 6. The principle shown by the blue color is based on pixels defined by the purple and red pixels, and the coordinates of the pixels fit the general circle equation. The second principle, shown by the red color, is characterized as an arc approximating a sinusoid function. In both principles, it is necessary to define the parameters to compensate for axis eccentricity and the angles between normal vectors.

The blue principle is based on the linear choice of pixels from the picture bounded by the annulus part of the tire. The transformation is based on lines of pixels in angular iteration (*φ_i_*). The general equation for the circle is defined in the Cartesian coordinate system.

The pixel positions are defined by the coordinate pixels *x_i_* and *y_i_*. The blue line in Figure 6 is definable in the polar coordinate system. Angle iteration (*φ_i_*) of the polar coordinate system characterizes the resolution in the *x*-axis, while *ρ* characterizes the resolution in the *y*-axis and the basic pixel coordination for *φ*_0_. Computing the angle iteration is a solvable method, as displayed in Figure 7. The main point involves defining the zero-point, that is, the start point of the circle in a polar coordination system. The zero-point is characterized by coordinates *x_m_* and *y_m_*, defined as the highest point. When adapted to the pixel coordinates in the picture, there exists a function to find a minimal position in the *x*-axis in the pixel edge polyline:
(1)$$x_{m}=x_{i},\;y_{m}=y_{i}=min\left(Y\right).$$

The next step allows for the definition of other features, such as the center of the partial annulus, defined by coordinates *x_t_* and *y_t_*, expressed in Equation (2), and angle iterations. The coordinates *x_i_* and *y_i_* represent the pixel edge polyline for every iteration of *φ_i_*, based on the main characteristics of circles and analyzing the points in the circle, according to its center point. The angle iteration is described in Equation (3), where the differences ∆*x* and ∆*y* are computed as differences from the zero-point and, from them, the angle iteration *φ_i_* is computed, as depicted in Figure 7.
(2)$$x_{t}=x_{m},\;y_{t}=y_{m}+\rho ,$$
(3)$$tan\;\varphi_{i}=\frac{∆y}{∆x},\;\begin{array}{c} ∆x=x_{m}-x_{i}\\ ∆y=y_{m}-y_{i} \end{array}.$$

The red principle, displayed in Figure 6, is described as choosing part of the circle with a specific radius, where pixel coordinates are defined in the *x*-axis with a specific radius. For every part of the circle (arc), the radius must be defined.

In the process of carrying out the polar transformation, it is necessary to choose an appropriate method. The blue method is, upon the first view, simpler than the red method; however, the blue method has complications involving defining the basic positions of pixels in the picture. In every case, it is necessary to adapt the system of Cartesian coordinates to pictures with the positions of pixels being closer to reality than that with polar coordinates. The following solution uses the red method. To gain relevant data, it is necessary to modify the data, as real data are noisy due to the captured geometric relief of the inspected objects. To address this, we conducted the smoothing of *φ_i_*, as defined in Equation (4). Modification of the result obtained from this equation was suitable, through the application of minimal and maximal functions, not to the first and last items in the angle iterations, but to find minimal and maximum values for the specific number of *φ_i_*. The coordinates in the *x*-axis are defined, from the picture, as every column of the picture matrix. It is necessary to calculate the pixel coordinates in the *y*-axis. Calculation of the radius is defined in Equation (5) for every iteration, where the number of iterations (n) defines the size of the picture, that is, 4112 × 3008 (x,y). Calculating the average radius ($\bar{\rho }$) in the numerator of Equation (5) was carried out at every iteration, where the final was a mean value of all radius iterations. In the radius analysis, the result had a large variance, due to the calculation style, as the input values were not continuous, but discrete values of pixel coordinates. This feature generates rounding, as demonstrated by the high variation of the results.
(4)$$\varphi_{i}∼min\left(\varphi_{i}\right)+\frac{\left(max\left(\varphi_{i}\right)-min\left(\varphi_{i}\right)\right)}{num\left(\varphi_{i}\right)}*i,$$
(5)$$\bar{\rho }≅\frac{\sum_{i=1}^{n}(\frac{∆x}{sin\left(\tan^{-1}\left(\frac{∆y}{∆x}\right)\right)})}{n},\;n=4112.$$

As a result, it is necessary to adapt to boundary features, such as edges and objects of interest. These edges are highlighted, in Figure 4, by red and purple curves, while the green curve represents the mean values of the red and purple curves. Calculating the radius values is inevitable when considering the mentioned boundary features. The expression of pixel positions in the *y*-axis can be performed in two ways: from the general circle equation or for the equation based on the principle of the polar transform. Modified equations for the captured picture are expressed in Figure 8, with Equation (6) in green and Equation (7) in blue, where *y_gi_* represents the y-coordinates when computing the green polyline and *y_bi_* represents the y-coordinates of blue polyline:(6)$$y_{gi}=-\sqrt{\rho^{2}-{\left(x_{i}-x_{m}\right)}^{2}}+y_{m}+\rho ,$$
(7)$$y_{bi}=\rho \left(1-\cos\varphi_{i}\right)+y_{m}.$$

The radius result for the red curve displayed in Figure 4, according to Equation (5), is 10,000, which represents the average value of all calculated radii at every iteration. A comparison of the calculated radii is illustrated in Figure 8 (left). There were significant errors in the computed green (6) and blue (7) curves, compared with the red curve, in the calculated radius values, as shown on the left side of Figure 8. Solving for the radius is possible through empirical or analytical methods, where empirical methods are based on a manually entered radius value with the main goal of approximating the computed curves to red curves as closely as possible. Modified radius values are demonstrated in Figure 8 (right), along with the declared radius values.

The differences in radius value were due to the computed curve coordinates, where both methods applied were based on different coordinate systems and equations; Equation (6) is based on the Cartesian coordinate system, while Equation (7) corresponds to a polar coordinate system adapted to the pixel coordinates in the picture. On the left side of Figure 9, apparent deviations in the *y*-axis can be seen, even after approximation of the curves to the target curve. The shapes of the curves were displaced in the *x*-axis, caused by the method for gaining the *x_m_* value, due to the uneven character of the curve. The manual corrections were carried out by displacing the *x_m_* value for (9), the renumbering of *φ_i_* by a value in (6), and renumbering of *φ_i_* by a value in (4). The results after such replacements are shown in Figure 9 (right), with the displacement values for specific curves. The method described above was similar to the standard method, and it was necessary to declare parameters as input data to perform the polar transform, mainly according to the type of picture (as illustrated in Figure 4), namely part of the annulus of any rotating object.

The second mentioned method is an analytical method. The analytical method was constructed using boundary conditions defined by the red and the purple curves displayed in Figure 4. According to these curves, it is possible to perform a polar transform on part of an annulus. Knowledge from an empirical method was applied in the design of the analytical method, with important parameters as displayed in Figure 10, where the main parameters were *ρ*_0_ (the red curve) and *ρ_j_* (the purple curve). Calculating radius parameters was possible, but it required mathematical expressions of the boundary curves. The approximation of curves was carried out using polynomial regression. The type of circle definition, selected as the most suitable, was second-degree polynomial regression, as shown in Equation (8), as it was considered closest to the circle equation.
(8)$$y_{ri}={\left(X^{T}X\right)}^{-1}X^{T}x_{i}=ax_{i}{}^{2}+bx_{i}+c.$$

This regression applies to both curves. From the polynomial expression of curves, we defined d*x*_0_ using the coordinates *x_m_* according to Equation (9) and *y_m_* according to Equation (10). This mathematical regression managed to center the computed curves to the boundary curves and no other displacement corrections were needed, such as in the empirical method, where the correction operation is required. The correction of radius values was performed using the result of Equation (8), adapting the radius value to the approximate curves obtained by Equations (6)–(8). This modification was carried out to determine the differences in the *y*-axis between curves described in Equation (11) for the green method, and Equation (13) for the blue method. Based on these equations, it was possible to express the exact value of the radius to approximate a regression polynomial equation with a defined boundary condition. The equations for the blue and the green methods expressed as Equations (12) and (14), were built on the condition of zero difference between the computed radius values and the polynomials. The application of the analytical method for the red curve is displayed in Figure 11, where it is compared to basic radius values derived from Equation (4) (the left side) and the computed radius values including the displacement parameters in the *y*-axis such as ∆d*y_gi_* (green), ∆d*y_bi_* (blue) and the equation of polynomial regression (*y_ri_*). The same principle, used for the purple curve, is displayed in Figure 12, with the same information as in Figure 11 on the right side. Figure 5 illustrates a possible camera placement, with respect to the captured object, which can cause deformation or distortion of the captured image by the camera system. These placement parameters emerged mainly in the green method. Displacement in the *x*-axis, as indicated in Figure 4 and Figure 5, is visualized as the displacement values in Figure 11 (right). The angle α resulted in different displacement values for the green method in Figure 11 (122 pixels) and Figure 12 (149 pixels). In the blue method, it is possible to see that there was a displacement difference, but it was lost by rounding to 4 decimal places. The angle ε manifested in radius values (Ro green), where the difference (1901) between the radii of the red (5467) and purple (3566) curves should have the same difference as for the *y_m_* values (1995) as for the red (337) and the purple (2332) curves.
(9)$$x_{m}=x_{i},\;\mathrm{where}\;\frac{\partial y_{ri}}{\partial x_{i}}=0$$
(10)$$y_{m}=\frac{\partial y_{ri}}{\partial x_{i}}\left(x_{m}\right)$$
(11)$$∆dy_{gi}=y_{ri}-y_{gi}=ax_{i}{}^{2}+bx_{i}+c-\left(-\sqrt{\rho^{2}-{\left(x_{i}-x_{m}\right)}^{2}}+y_{m}+\rho \right),\;i\in \langle 1,4112\rangle$$
(12)$$\mathrm{if}\;∆dy_{gi}=0,\;\rho =\frac{ax_{i}{}^{2}+bx_{i}+c-y_{m}+{\left(x_{i}-x_{m}\right)}^{2}}{2\left(ax_{i}{}^{2}+bx_{i}+c-y_{m}\right)}$$
(13)$$∆dy_{bi}=\;y_{\mathrm{ri}}-y_{bi}=ax_{i}{}^{2}+bx_{i}+c-\left(\rho \left(1-\cos{\;\varphi }_{i}\right)+y_{m}\right),\;i\in \langle 1,4112\rangle$$
(14)$$\mathrm{if}\;∆dy_{bi}=0,\;\rho =\frac{ax_{i}{}^{2}+bx_{i}+c-y_{m}}{\left(1-\cos\;\varphi_{i}\right)}$$

The methods described above allowed us to obtain the boundary parameters necessary to perform polar transformation when considering non-well-placed camera hardware. Based on the above-mentioned aspects, the analytical method based on the green principle defined in the Cartesian coordinate system was chosen. Figure 10 defines another variable, *X_k_*, as a matrix of pixel coordinates in the *x*-axis. The index *k* represents the row in the picture, for which the start is defined as *y_m_* in the red curve (*y*_0_) and the end as *y_m_* in the purple curve (*y_j_*). Under this principle, we computed various *k*-variables, including the radius *ρ_k_* (15), ∆d*x_k_* (16), *n_k_* (17), *X_k_* (18), and *Y_k_* (19) in the *k*th iteration. Together, they represent a gradual transition from the red to purple parameters. The parameter *n_k_* defines the number of pixels in a row, or how many pixels will be used from the basic picture. Excluding the *n_k_* value can lead to deformation in the transformed picture. In Figure 10, we illustrate a shorter purple curve than the red curve. The parameter *n_k_* representing the shortening of the transforming curve was computed based on the boundary parameters of radius values, as defined in Equation (17). This value, along with the ∆d*x_k_* value for displacement correction, expresses *X_k_* as indices in the *x*-axis of pixels. Coordinates in the *y*-axis, *Y_k_*, were computed by Equation (19). The results used for the above-mentioned green analytical method in the polar transformation are displayed in Figure 13, where the original image is shown on the left side and the transformed picture is on the right side. The inequalities in the transformed picture are negligible, and the final picture appears to be linear.
(15)$$\rho_{k}=\rho_{0}-\left(\rho_{j}-\rho_{0}\right)\frac{k}{y_{j}-y_{0}},\;k\in \langle 0,\left(y_{j}-y_{0}\right)\rangle$$
(16)$$∆dx_{k}=dx_{0}-\left(dx_{j}-dx_{0}\right)\frac{k}{y_{j}-y_{0}},\;k\in \langle 0,\left(y_{j}-y_{0}\right)\rangle$$
(17)$$n_{k}=n_{0}-\left(1-\left(\frac{\rho_{0}-\rho_{k}}{\rho_{0}}\right)\left(\frac{k}{y_{j}-y_{0}}\right)\right),\;k\in \langle 0,\left(y_{j}-y_{0}\right)\rangle$$
(18)$$X_{k}=\langle \left(\frac{n}{2}+∆dx_{k}-\frac{n_{k}}{2}\right),\;\left(\frac{n}{2}+∆dx_{k}+\frac{n_{k}}{2}\right)\rangle ,\;k\in \langle 0,\left(y_{j}-y_{0}\right)\rangle ,\;n=4112$$
(19)$$Y_{k}=-\sqrt{\rho^{2}-{\left(X_{k}-x_{m}\right)}^{2}}+y_{m}+\rho$$

### 2.2. Point Cloud from the Laser Sensor

Previously published works have described the principles of working with point clouds obtained by a laser sensor [34] to obtain visual content compatible with algorithms designed for camera-obtained imagery [3,31]. The further work described in this paper is based on the results of these works. The described procedures to generate visual content from point clouds used data consisting of 12 scans, with the shape of 25,000 × 640, representing a specific part of the tire sidewall area. It focuses on the connection of specific parts of the scan, in order to obtain a scan of the whole tire sidewall. The first principle is based on finding matching geometrical data between scans, as performed by matrix matching in a stepwise manner (design of laser scanner data processing and their use in the visual inspection system, ICIE2020). This matching was accurate, but the algorithm is computationally intensive. Such a procedure took approximately 10 min on an Intel Core i7-8700. Better results were obtained by considering the pattern recognition of visual content [2], where the convolution principle was used through the function MatchTemplate, integrated into the OpenCV library [35]. The task time was reduced to 3 s, including GUI procedures, using similar hardware as in the case of matrix matching. Further work has utilized the pattern recognition method for two main reasons: the speed of the procedure and the possibility to recognize pre-defined patterns [36,37,38]. For this purpose, a database of pre-defined patterns, composed of letters and other features regularly occurring on tire sidewalls (as mentioned in Table 1), was created. This database contains these data:Coordinates of the snipped pattern;Pattern from the grayscale image generated from the point cloud;Point cloud of the pattern;Positions in string chain;Pattern from a color image converted from grayscale to color.

This database was used to identify the basic patterns and basic objects identifiable on the tire surface, as illustrated in Figure 14. The main advantage is its fast application and low computational intensity when compared to a CNN. This procedure can save time and computational power, which is a necessary aspect of inspection tasks for mass production under factory conditions. According to obtained data, it is possible to assume the position of other patterns or to correlate the positions of recognized patterns with pre-defined patterns, where irregularities between defined patterns and inspected areas indicate the possibility of an abnormality, defect, or misidentified object occurring.

The ability to use pattern recognition is conditioned by the homogeneity of the data, where any variance leads to an inability to recognize patterns; for instance, this method is not frequently used when analyzing visual content from a camera, as the variance in pictures containing the same objects is typically very high. The deployment of this method in the camera vision field relies on strict adherence to conditions during capturing, such as light conditions, position, and so on. The difference in our approach lies in the application of pictures generated from laser sensors, thus. being based on scanned surfaces. In contrast, the variance in these data is low, which is computed as subtracting the matched area of the first scan from the second scan. Conditions were normalized matrices and filtered values, mainly in missing data. In matched scans (e.g., Scan 1 and Scan 3 mentioned in [2]), the differences mainly ranged between −0.2 and 0.2 mm, as displayed in Figure 15. There were occurrences of larger absolute differences, almost up to 3 mm, in which the number of these values corresponded with the number of differences to 0.4 mm. Therefore, an absolute difference in the range between 0.4 and 3 mm likely indicates a defect occurring on the scanned surface.

Based on the above, it is possible to design an inspection system that is capable of detecting defects occurring on the scanned surfaces of the inspected objects. The principle is based on the definition of the correct data, denoted as the standard reference data, which represents correctly scanned objects without any defect or abnormality. Such a system is applicable in cases where the final product has a homogenous character. A particularly suitable industry for the deployment of such an inspection system is the production of printed circuit boards (PCBs) or similar objects. In the case of soft materials, such as polyurethane or rubber, it is more complicated, due to slight changes in the shapes of objects and their features. Despite non-ideal properties, we were able to use the comparative system to evaluate tires. The results are displayed in Figure 16, where three abnormalities were found. The abnormalities were all observed as deviations higher than 0.5 mm in absolute value. To process such data, it is necessary to use value clustering algorithms. Appropriate clustering algorithms are part of unsupervised learning methods in the field of artificial intelligence. The appropriate type of clustering depends on parameters, which can limit the applicability of a given method. One of these is of very high importance, especially in mass production: the time requirements. Another is the ability to separate different values into specific numbers that correspond to the number of defects occurring in the scanned object. In terms of time consumed, the best method is K-means [39,40]. Another possibility is the DBSCAN algorithm, which is a little slower (mainly when considering huge data sets) but can separate point clouds into clusters without defining the number of clusters first [41,42]. As displayed in Figure 16, this inspection system can detect abnormalities or defects. The figure shows three clusters, while more clusters were found in the data; however, they were too small and, therefore, irrelevant. Furthermore, the threshold for categorizing clusters as important or irrelevant depends on the expectation of the user of the system. A special case is Abnormality 4, an item occurring in both scans. Recognition of this abnormality was based on the way of scanning, where a geometrically diverse object is represented by small scan values but has large enough differences to have them categorized as abnormalities.

Figure 16 shows abnormalities in 2D generated from a point cloud (i.e., 3D data). Based on this, it is possible to visualize abnormalities in 3D, as shown in Figure 17, using the Open3D library [43]. The time necessary to perform clustering ranges from 1 to 3 s, depending on the number of differences larger than 0.5 mm, which, in Scan 2, was 74,507 different points. Points are shown in the colors green and red, where green indicates points within the tolerances regarding the standard reference data, while red highlights points that are out of the tolerance band. Furthermore, it is possible to see the real surface captured by the laser sensor. This feature is mainly interesting for personnel outside of the company, who are able to see the surface defects in real-time. The 3D data can offer more possibilities for analyzing defects, while the use of 2D data may complicate the understanding of the character of defects, as they may not be obvious. Only visual and geometric defects, such as missing or excess material, can be detected [3]. The analysis of defects and manufacturing processes can reveal the origin of the defect, leading to modification of the processes in such a way that the frequency and severity of defects due to the problems in the production process can be decreased. Such an implementation would also help to better manage product quality.

### 2.3. Fusion Geometric Data and Pictures from the Camera

Section 2.1 and Section 2.3 described the data captured from the tire sidewall using a camera (2.1) and by a laser sensor (2.2). Each of these methods has specific advantages and disadvantages. As such, it is possible to obtain a synergic effect by combining these data and, thus, by combining the advantages of individual methods. The fusion of this data is shown in Figure 18, which compares individual and merged pictures (merged data). To merge data, it was necessary to normalize the data types and obtain the same shape. For this reason, we developed an analytical method of unfolding, where the main aim was to suppress and minimize the inaccuracies caused by hardware depicted in Figure 5. In the case when we did not apply this method, blurred areas occurred in the merged data, mainly at edges, such as the borders of letters, symbols, and the tire tread. The blurring was caused by non-corresponding object positions, such as edges in different places in the visual and geometric data. To perform the merging process, it was necessary to unify the size of the fused data through resizing. Resizing the smaller data types and unifying the dimension to the larger image offered better results. Otherwise, the resolution would be lost, in the case of pictures with higher resolution. To fuse the data, it was necessary to define merging points, such as defined samples recognized in the captured data.

### 2.4. Defect Detection by RCNN with VGG-16 Network

According to previous works [18,23,44], existing defect detection methods have mainly been based on deep learning (supervised learning) using specifically designed or pre-trained CNN architecture, such as AlexNet [45], Resnet-50 [46], or VGG-16 [18]. For this reason, evaluations have been performed using these methods on specific data, such as visual content generated from laser sensor data [2]. We chose to test the VGG-16 CNN architecture, based on results presented in [18,47,48]. Training was performed on 5000 samples, including two categories of defects: Impurity with the same material as the tire material (CH01), and mechanical damage to integrity (CH05). These defects were defined as the simplest to detect, due to their size and visibility in the data. Training was performed using MATLAB software. The training options were set as follows: Stochastic Gradient Descent with momentum (sgdm), minibatch size = 16, maxepochs = 5, initial learn rate = 0.000001, and execution environment = parallel. After training, the detector reached an accuracy of 93.75%. The detection was performed on 11 scans [2], where 10 detected possible defects with the highest accuracies. In this way, there were many incorrect detections. For instance, in Figure 19, we show the result of detection based on VGG-16 in SCAN 1, where four areas were detected to have a rubber impurity defect (CH1), but the areas were not bounded very well. The same results were observed for every scan. The other important parameter was time to perform detection which, for one scan, ranged approximately from 100 to 125 s (i.e., SCAN 1, 116.258 s). The hardware used for training and detection was a GPU (Intel Core i7-8700, RAM 32 GB DDR4, NVIDIA GeForce RTX 2070). The size of the scan was 640 × 25,000 pixels (16 MPX). The results of this experiment appeared not to be very promising, according to the long time required for detection and incorrect detections while declaring high accuracy.

### 2.5. Classification of Detected Abnormalities

Based on the results presented in Section 2.4, we decided to design a new VGG-16 model; however, not for performing the detection of defects, but, instead, detecting the classification of defects from the data set. The model training was performed on 4000 samples, including two types of defects (CH1 and CH5). The validation data set contained 1000 samples. Training settings were as follows: Optimizer = ADAM, batch size = 96, and epochs = 20. Training was performed using KERAS. All samples were resized to 64 × 64 pixels. The process of training is displayed in Figure 20. The evaluation of the trained model is depicted in Figure 21, as a confusion matrix.

The trained model was used for the classification of abnormalities obtained from the process described in Section 2.2, where three abnormalities were detected (Figure 16). The difference between “abnormalities” and “defects” is that an abnormality is detected by an unsupervised method (DBSCAN) as something different, but without other information. In real conditions, an abnormality should be labeled as a specific defect. To define and classify recognized abnormalities, it is necessary to perform the classification of abnormalities; for example, by training a CNN. In this case, VGG-16 was used. The classification of abnormalities was performed, where Abnormality 7 and Abnormality 2 were classified as rubber impurity (CH1) (Figure 22), and Abnormality 4 was classified as mechanical damage to integrity (CH5). The accuracy of classification for Abnormality 7 as rubber impurity was 73.11% in the graph (Figure 22; left), corresponding to the label “0”. The time necessary for the classification of one item was approximately one second, using the same hardware as mentioned above.

## 3. Results

In Section 2, we described five main methods. The first involved processing visual data captured by a camera (2.1 Camera vision). The second involved processing data captured by a laser sensor (2.2 Point cloud from laser sensor). The third involved fusing the data obtained from the above. The fourth involved the use of a deep learning method (i.e., VGG-16) for defect detection in visual data from the laser sensor as part of tire inspection. The final section described the application of VGG-16 for the classification of abnormalities obtained by the process described in Section 2.3. In the Introduction, we posed four hypotheses answered in the following:

**Hypothesis** **1** **(H1).**
*Yes, it is possible to automate the polar transform procedure of the captured partial tire sidewall using a system based on polynomial regression.*


**Hypothesis** **2** **(H2).**
*Yes, it is possible to compensate for the inaccuracies in visual data captured by a camera, with a minimum of two boundary conditions, through the polynomial expression of the detected edges. The concept is illustrated in the Ro green values corresponding to the Cartesian coordinate system, where Ro green for the red curve was 5467, and for the purple curve was 3566. The positions of the curves were $dx_{0}=337,\;dx_{j}=2332$ (Figure 10), where the difference in Ro values was 1901 and the difference in position of the curves was 1995, which represents compensation for the inappropriately set up hardware.*


**Hypothesis** **3** **(H3).**
*In the Material and Methods section, we applied a deep learning method, that is, an R-CNN based on the VGG-16 network. The results for the visual data generated from the laser sensors showed little potential for application.*


**Hypothesis** **4** **(H4).**
*We described a tire inspection system using unsupervised learning (in particular, the DBSCAN algorithm), which showed good potential for application. The main advantage lies in detecting abnormalities, which can be further classified as defects or acceptable items.*


An overview of Section 2 is provided in Figure 23, illustrating the principle of the designed tire inspection system based on unsupervised learning; specifically, the DBSCAN algorithm. The sensor devices are the camera system and the laser sensor.

In the Materials and Methods, we described the processing of the camera-obtained data using a method for unfolding the specific area of pictures capturing the tire sidewall. The resolution of the captured pictures was 4112 × 3008 pixels. The first unfolding method used a picture of the whole tire sidewall, and the unfolding was divided into two steps. During the first step, circles in the picture were detected. The second step was a polar transformation that used the parameters of the circles obtained in the first step. The result of this processing is shown in Figure 3. The tire sidewalls were usually not fully straight, but, instead, slightly wavy. The harmonic analysis indicated the occurrence of a non-zero first harmonic component, as caused by the eccentricity of the detected circle when compared to the real center point of the tire sidewall. Circle detection was carried out using Hough circles implemented in the OpenCV library. For the polar transformation and conversion to the Cartesian system, the polarTransform library was used. The hardware used for computing was as follows: Intel Core i7-8700, RAM 32 GB DDR4, and NVIDIA GeForce RTX 2070. The whole process took less than 3 s. The most notable disadvantage of this unfolding process is setting up the parameters for circle detection and polar transformation. For this, the empirical setting of the mentioned parameters is necessary and, even still, some eccentricity persists, which can be further removed by the manual centering of the picture before the transformation. The size of Figure 3 is 8224 × 791 pixels. To increase the resolution, only part of the tire sidewall was considered at the same time, as shown in Figure 4. The complications with setting up the empirical parameters were the same as in Figure 2. To solve this, a method constructed using boundary conditions based on edge detection to define the part of an annulus was designed, as displayed in Figure 4. Another possible way to perform polar transform from empirical to analytical is by using second-order polynomial regression, as mentioned above. The main advantage of this lies in the possibility of automating the definition of parameters and compensation for inaccuracies of the camera system (as described in Figure 5). This analytical method is based on regression, and it seems to allow for the automation of unfolding pictures. For this reason, it was chosen for use in the tire inspection system.

The second subsection described processing 3D geometrical data, captured by a laser sensor, into 2D data, as depicted in Figure 14. The results of pattern recognition with respect to pre-defined samples are described in Table 1. Pattern recognition was closely described for processing 3D data from laser sensors into visual content [2], for which we used the function cv2.matchTemplate integrated into the OpenCV library. The condition for using this method is that the data must be homogenous, as illustrated in Figure 15. In the case of conventional images, the implementation of pattern recognition is complicated by the high dispersion caused by the light conditions and requires the strict positioning of the object during the process of capturing images. The other advantages of using geometrical data for the tire inspection system are explicitly defining the correct areas and the high possibility of capturing geometrical abnormalities, as displayed in Figure 16. Abnormalities were classified as clusters, separated by the unsupervised learning algorithm DBSCAN, which is able to separate data into a specific number of clusters based on density without a pre-defined number of clusters. Data chosen for clustering were defined based on differences in 3D data; specifically, in the z-matrix, where the threshold was defined as 0.5 mm, according to the data displayed in Figure 15. The detected abnormalities can be described in 3D, as displayed in Figure 17. The classification of detected abnormalities is the subject of Section 2.5, where the VGG-16 network architecture was utilized by means of KERAS in Python. The detected abnormalities were correctly classified as rubber impurity (CH1) for Abnormalities 2 and 7, and mechanical damage to integrity (CH5) for Abnormality 4. Additionally, we described the use of the R-CNN method for defect detection, performed in MATLAB, with the goal to design a tire inspection system based only on supervised learning. The results of this experiment did not show much promise, and the design of the model (neural network) needed much improvement. In Section 2.3, we described the fusion of the captured data (by camera and laser). Fusion was performed by manual matching and resizing data to uniform size. Centering was based on the upper left rectangular corner of the specified area, representing the same object pattern in both types of data.

An overview of the constructed tire inspection system is shown in Figure 18, where the described methods include the camera image polar transform based on polynomial regression (supervised learning). For the data from the laser sensor, the visual content generated from geometric data is used, pattern recognition of pre-defined samples by cv2.matchTemplate is carried out, the detection of abnormalities is performed by the DBSCAN algorithm (unsupervised learning), and the classification of abnormalities to defects relies on the VGG-16 network architecture (deep learning).

## 4. Discussion

As mentioned in the results, Section 2 was divided into five subsections. The first was focused on the polar transform for the unfolding of the part of the annulus, capturing part of the tire sidewall. We covered the conventional method of unfolding by circle Hough transform and polar transform to the Cartesian coordinate system. The disadvantage of this method is the non-concentricity of the detected circle and the real tire sidewall, which manifested as the first harmonic component shown in Figure 3. The next step was the development and description of a method for the polar transform of the part of the sidewall based on boundary conditions. The boundary conditions were set as the detected edges describing an area of the captured surface. Edge detection is sensitive to the characteristics of the original captured picture. This paper presents the use of picture modification to suppress the background of the original picture. In a fully automated implementation of the process of capturing pictures, the camera should be enhanced by optimizing the background of the tire inspection stand: the background should allow for clear separation of the tire sidewall from the background. Figure 5 shows inaccuracies in the capturing of tire sidewalls when using a camera. The eccentricity of the ε angle causes the projection of the captured circle as an ellipse, deforming the captured shape. For this reason, a more accurate method could modify the circle equation to that of an ellipse by adding appropriate parameters. In the case of a very low angle, the impact is negligible. Using polynomial regression, the parameters of the ellipse, compensated by values “a” and “b,” are shown in Equation (8). Therefore, we can perform unfolding while compensating for the inaccuracies caused by inappropriate hardware conditions, which can be suppressed as part of the calibration process. This polar transform process, which suppresses inaccuracies, is appropriate as part of tire inspection stand calibration, where the computed parameters could be set up as constants and applied in the process of the automated tire inspection system.

The second subsection focused on the geometrical data. The main aim was to perform pattern recognition, using cv2.matchTemplate, in compensation for the disadvantages associated with the conventional use of CNN. The common feature of both mentioned approaches is convolution. However, the implementation of pattern recognition is simpler than CNN (deep learning) and, thus, is more effective from a time–cost point of view. Using pattern recognition in conventional pictures captured by cameras is almost useless, due to the condition of capturing images and the character of the examined object. In the case of pictures generated from 3D data, the images are more stable and, therefore, more suitable to perform pattern recognition using the above-mentioned function. The other advantage is the possibility of processing 3D data from the laser sensor in the same way as 2D when a picture is generated from a point cloud. The detected abnormalities displayed in Figure 16 are displayed in 3D in Figure 17. We determined that using the DBSCAN method (an unsupervised learning approach) provides an effective means to detect abnormality clusters. The main advantage of this method is automatic clustering to a specific number of clusters; however, the disadvantage is its time performance. Based on the fact that DBSCAN clustering is performed only on the part of data characterized by differences larger than 0.5 mm in the *z*-axis, the time necessary to perform such a process is proportional to the number of differences. The third subsection was focused on merging 3D and 2D data. The main advantage was obtaining both the geometric and visual character of the scanned surface. According to this feature, it is possible to describe the abnormalities as a geometric feature, as well as a visual feature, without defining its character or category. The classification of abnormalities was described in Section 2.5. Very good classification results were observed, as shown in Figure 21. Using classification is necessary, as clustering algorithms alone are generally unable to categorize the character of abnormalities. We also described using an R-CNN-based approach to detect defects; unfortunately, the results were not very good. The declared detection accuracies were high, but the detected regions of defects were not appropriate or in the right place, as illustrated in Figure 19. In the introduction, we mentioned various categories of defects, which are illustrated in Figure 1. In real conditions, defects occur with many variations in shape, position, orientation, and so on. For this reason, is it appropriate to use only supervised methods for defect detection? Is it possible for a trained system to detect untrained objects? We found that the proposed tire inspection system can resolve this situation, by using an unsupervised learning approach—specifically, the DBSCAN algorithm—to separate points into clusters and detect every detectable abnormality in a specific place, according to the quality of obtained data. The issue of size in defect detection was suppressed in the tire inspection system’s classification procedure, where the input layer was declared to have a strict shape, and every abnormality was accordingly resized to perform classification. To ensure high defect detection accuracy in many orientations, it is appropriate to include these samples into the data set intended for training the CNN; however, there is still a high possibility of correct classification, due to the affinity of a given abnormality to its associated category.

In the Ph.D. thesis [4], tire inspection was performed, where the defect detection accuracy was in the range of 85.15% (CH04) to 99.34% (CH03). It should be noted that the accuracy of classification for Abnormality 7 as rubber impurity was 73.11% (Figure 22), less than that achieved by the conventional method described in the Ph.D. thesis.

## 5. Conclusions

In this paper, we described the design of a hybrid tire inspection system utilizing both 3D and 2D data. The applied algorithm combines both supervised and unsupervised learning methods. In terms of supervised learning, it uses pattern recognition and polynomial regression, while, in terms of unsupervised learning, the DBSCAN algorithm is used for the clustering task. Polynomial regression was used to automate the process compensating for the inaccuracies described in Figure 5, thus, replacing the conventional methods described in Section 2. Further work should involve managing and modifying the tire inspection stand and design background in order to identify and possibly automatically separate areas of the tire sidewall from the background. Another possible improvement would be to replace the circle with an ellipse in the relevant equation provided in Section 2. To capture the visual character of the surface, a conventional camera was used. The polar transform of images was necessary in order to allow for its unification with the data obtained from a laser sensor. In further work, the conventional camera may be replaced by a line-scan camera, which would allow for obtaining a much higher resolution and the same character of the captured surface as that derived from the laser sensor. The reason that we used a conventional camera was due to availability and, possibly, the use of color too. In the case of using a color line camera, the situation is more complicated, due to their availability on the market—such a type of line camera is quite rare.

Considering the laser sensor, this feature greatly improves the possibilities of hybrid tire inspection systems, based mainly on the use of geometric data. The geometric data reflect the real topology of the scanned surface and the much higher resistance of the laser sensor to the light conditions means that the data obtained from the laser sensor are more stable when compared to data obtained using a standard camera. Laser sensors can work as line cameras but, in comparison to conventional cameras, the resolution is lower, and they offer only grayscale images captured solely in the wavelength of the laser used. The application of a camera allows for the use of devices that operate on color images. In the introduction, we mentioned the use of CNNs for defect detection in wafers in the semiconductor industry. There is very important information in conventional inspection systems that works well when applying deep learning methods. The results of these systems are generally very good, achieving 94% and higher defect-recognition accuracy. However, there is also a need to manage larger data sets in order to train such models. Deep learning methods are in the category of supervised learning, as part of artificial intelligence. This means that defect detection is achieved through the use of a training data set and manual labeling of the defects. Therefore, in the case of very specific or abnormal defects occurring, the supervised system may not be able to capture the defect due to the absence of training data guiding the model to detect these types of defects. The proposed method works on 3D data and uses an unsupervised learning approach—a specific DBSCAN method—to identify abnormalities without the need to train or adapt tire inspection systems to capture such defects. In the future, it will be necessary to implement the described tire inspection system on a larger scale for every conceivable type of defect, including items such as barcodes and labels, and to perform verification of the defect. The system described in this paper can, in the future, replace conventional methods in inspection systems; however, it is still necessary to fulfill the mentioned aspects missing from this inspection system and verify its use in other industrial fields; for example, in the semiconductor industry, for such tasks as detecting defect on PCB wafers. Furthermore, in the future, exploration of the possibility of evaluating the angles α and ε mentioned in Figure 5 should be considered, leading to the possibility of designing a calibration system for the camera to obtain higher accuracy.

## Figures and Tables

**Figure 1 sensors-21-07073-f001:**
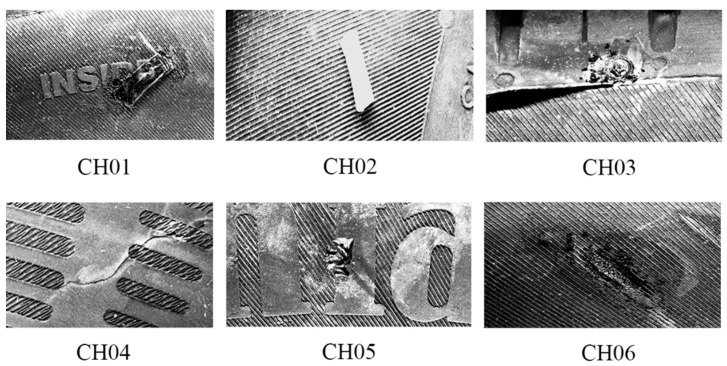
Samples of the most frequent defects [4].

**Figure 2 sensors-21-07073-f002:**
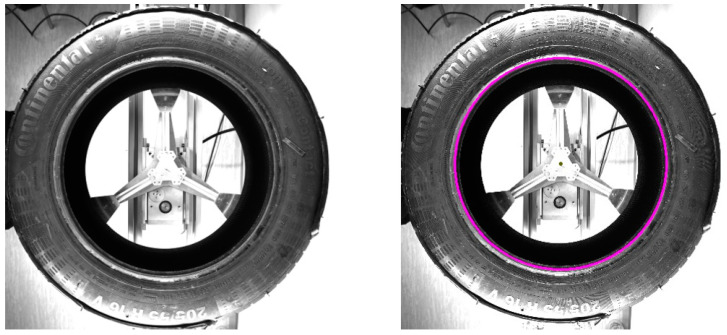
The sidewall of the tire.

**Figure 3 sensors-21-07073-f003:**
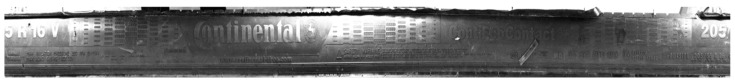
Unfolded sidewall image.

**Figure 4 sensors-21-07073-f004:**
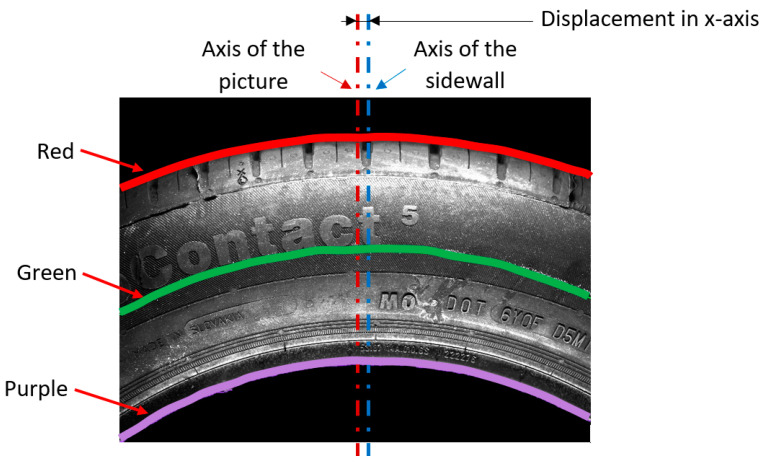
Part of the sidewall.

**Figure 5 sensors-21-07073-f005:**
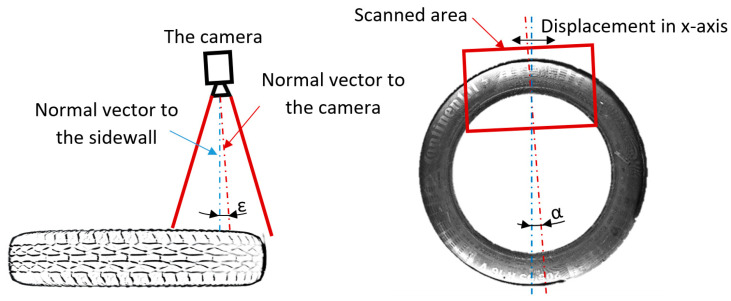
Illustration of eccentricity and angle when capturing part of the sidewall.

**Figure 6 sensors-21-07073-f006:**
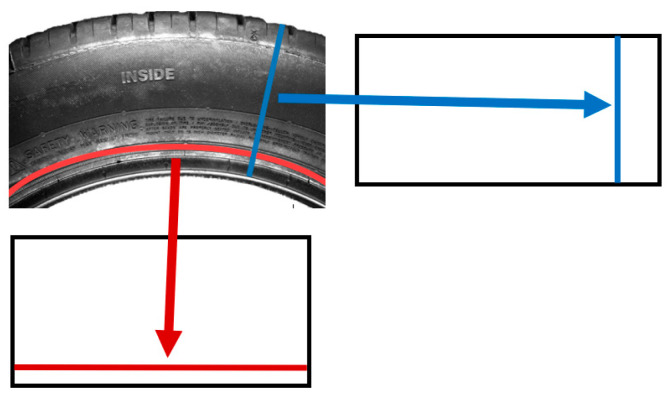
Possible unfolding of the sidewall area.

**Figure 7 sensors-21-07073-f007:**
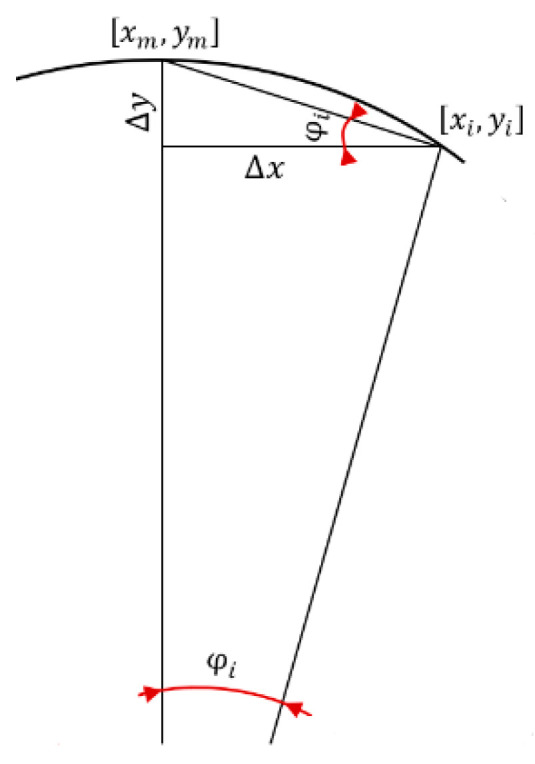
Scheme for the definition of *φi*.

**Figure 8 sensors-21-07073-f008:**
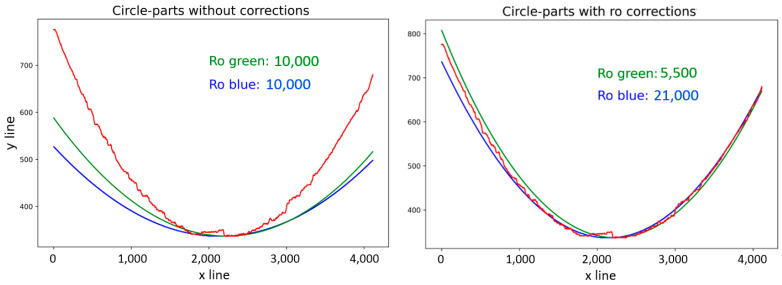
Circle part approximations, with radius (Ro) values.

**Figure 9 sensors-21-07073-f009:**
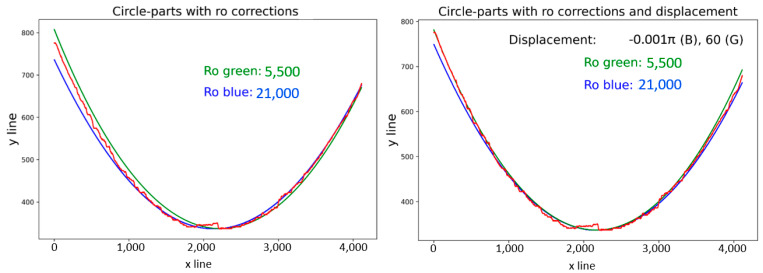
Application of displacement corrections.

**Figure 10 sensors-21-07073-f010:**
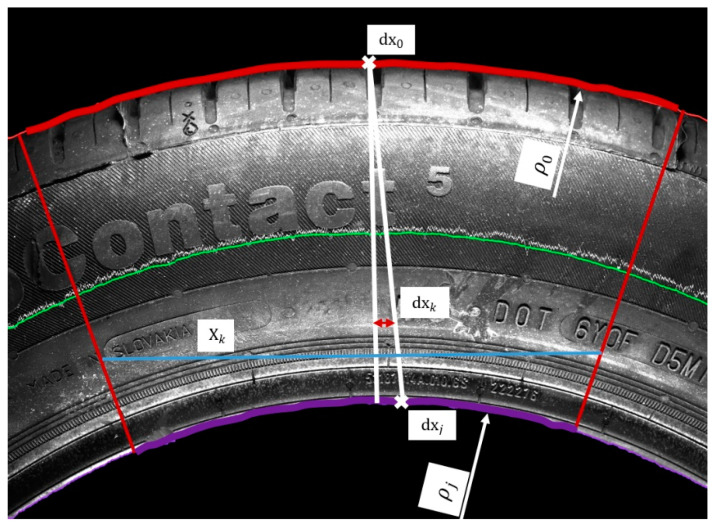
Basic principle and parameters of the proposed analytical method.

**Figure 11 sensors-21-07073-f011:**
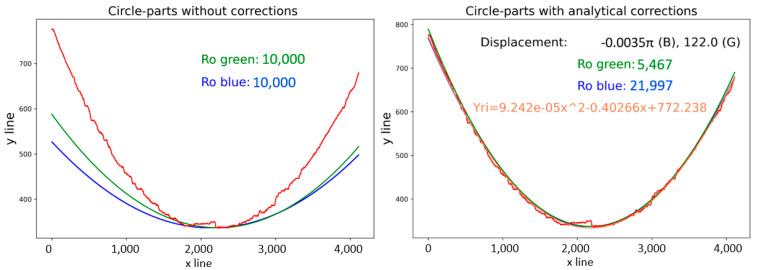
Comparison of computed radius and modified radius values by analytical method for the red curve.

**Figure 12 sensors-21-07073-f012:**
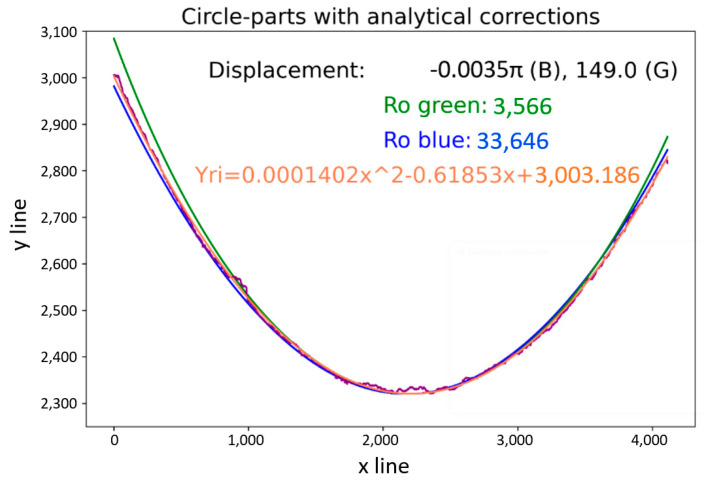
Radius values modified by the analytical method for the purple curve.

**Figure 13 sensors-21-07073-f013:**
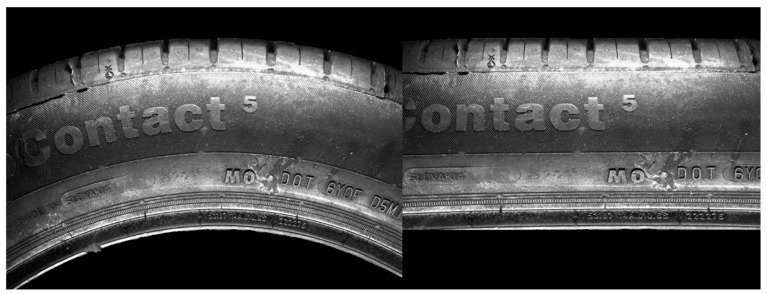
Result of polar transformation by analytical method.

**Figure 14 sensors-21-07073-f014:**
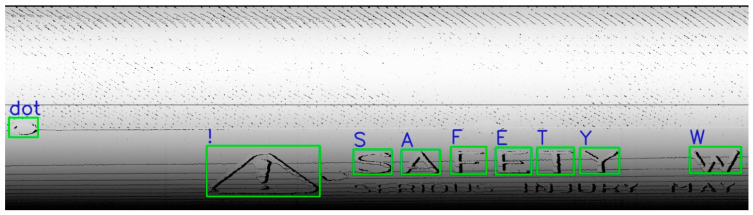
Application of recognizing pre-defined patterns.

**Figure 15 sensors-21-07073-f015:**
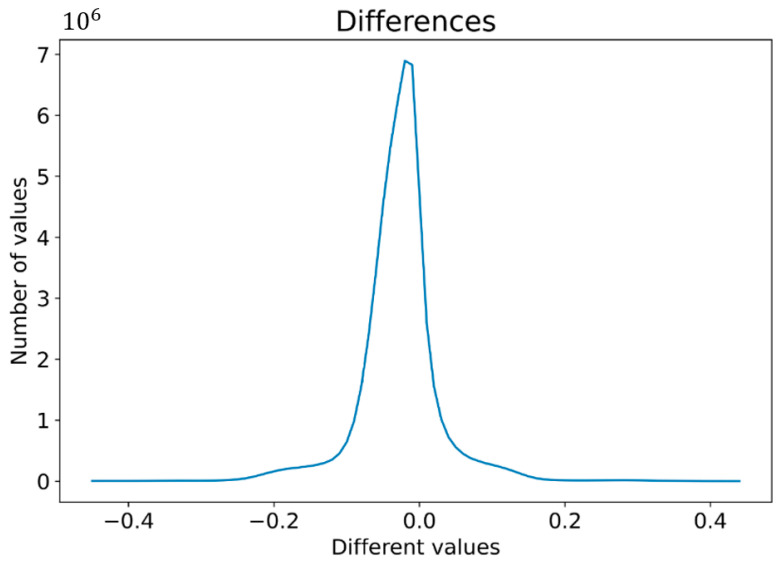
Different values in matched areas of scans.

**Figure 16 sensors-21-07073-f016:**
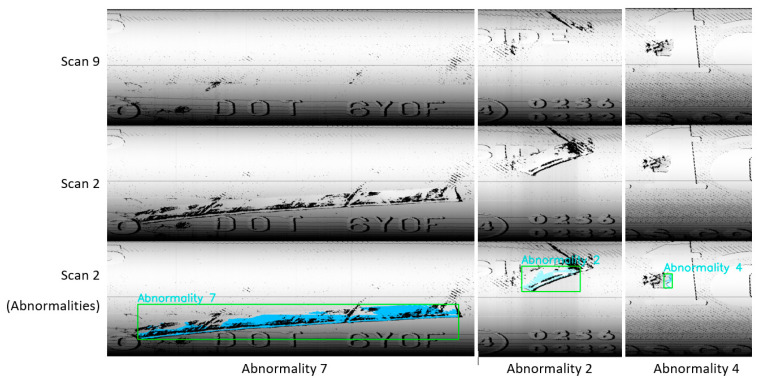
Detected abnormalities.

**Figure 17 sensors-21-07073-f017:**
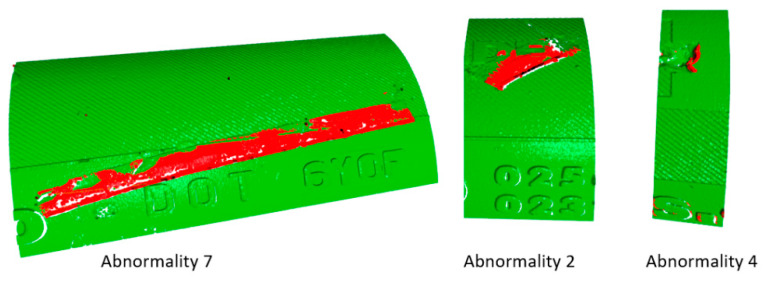
3D visualization of detected abnormalities.

**Figure 18 sensors-21-07073-f018:**
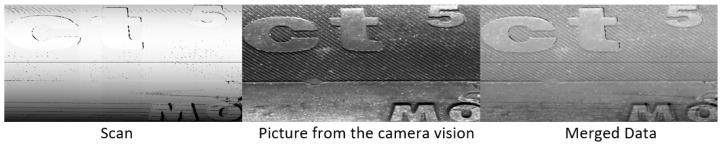
Visualization of merged data types.

**Figure 19 sensors-21-07073-f019:**
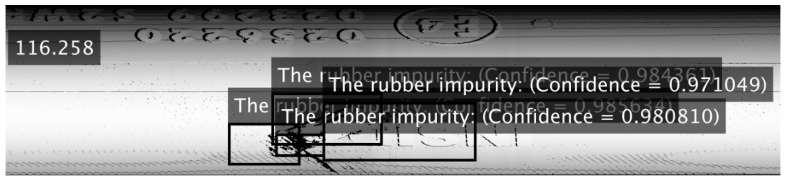
Defect detection by VGG-16 in SCAN 1.

**Figure 20 sensors-21-07073-f020:**
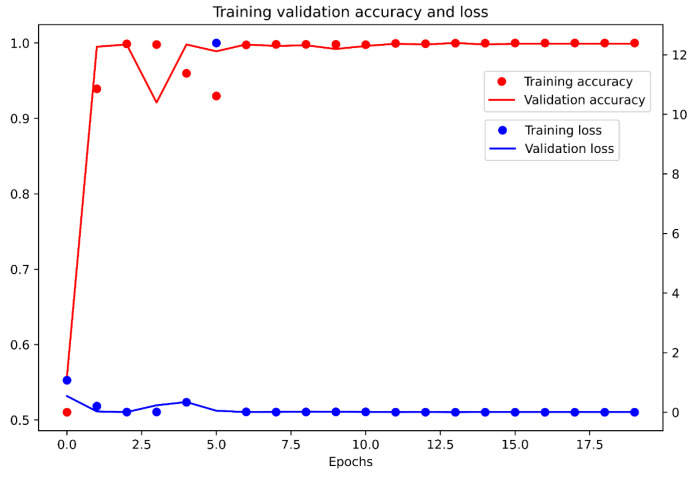
Process of training VGG-16 for classification.

**Figure 21 sensors-21-07073-f021:**
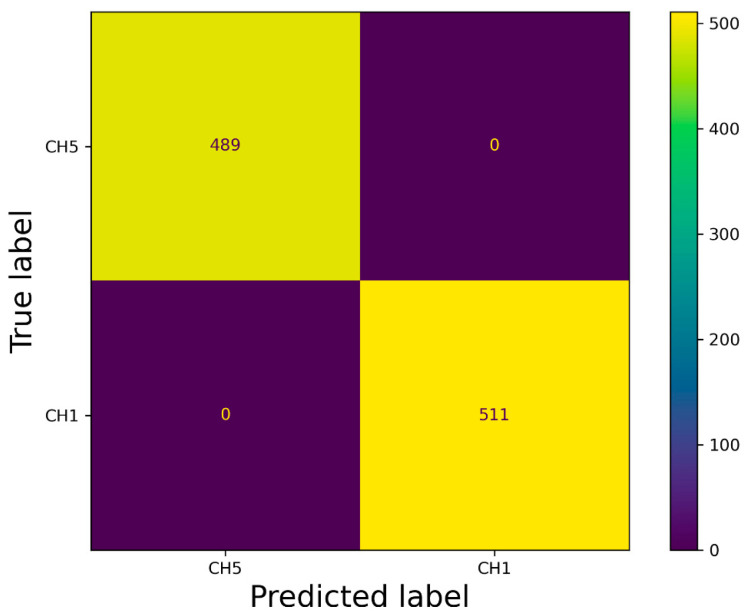
Confusion matrix for classification by VGG-16 network.

**Figure 22 sensors-21-07073-f022:**
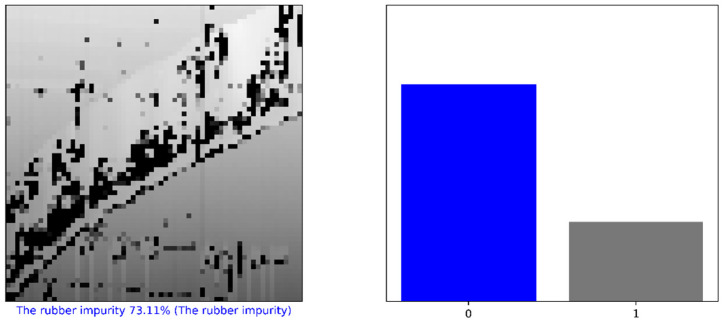
Classification of Abnormality 7.

**Figure 23 sensors-21-07073-f023:**
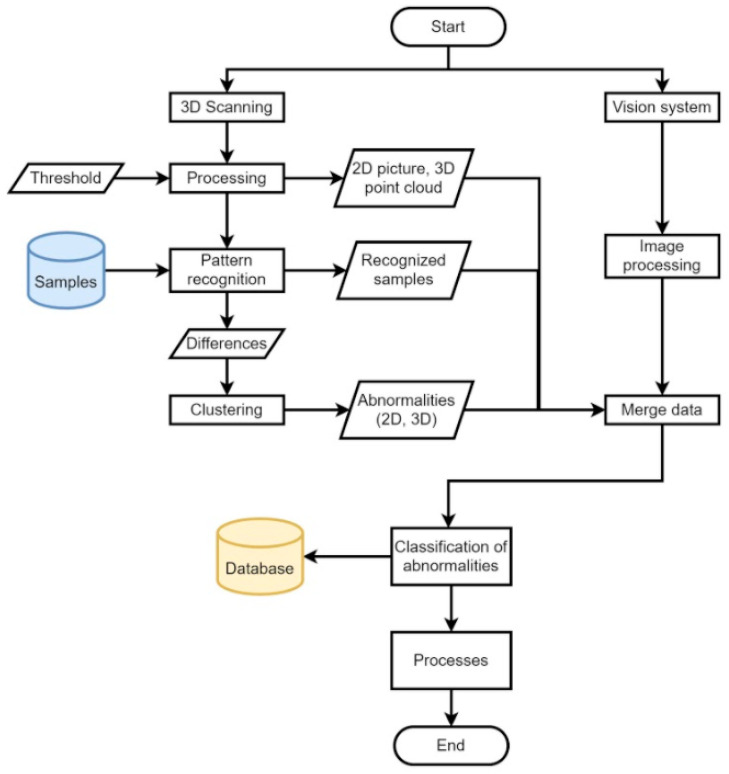
Overall design of proposed tire inspection system.

**Table 1 sensors-21-07073-t001:** Number of samples for specific scans.

Number of Scans	Number of Samples
Scan 1	107
Scan 2	126
Scan 4	91

## Data Availability

The data presented in this study are available on request from the corresponding author.

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
