# Peer review of "Analysis of the Possibilities of Tire-Defect Inspection Based on Unsupervised Learning and Deep Learning"

_sensors, 2021, doi:10.3390/s21217073_

Round 1

Reviewer 1 Report

Following the analysis of the work after the first review, I maintain my opinions from the first report. The authors did not make any changes to respond to the comments made.

Author Response

Dear Reviewer 1.

Thanks for your comments to the manuscript. It was performed changes including the professional language editing by MDPI in area of Computer science and Mathematics. To your comments from first review was noted in enclose word the biggest changes. After a few revisions, the are many changes in this manuscript.  
Thanks for our colaborating and your revision of the manuscript.
Yours sincerelly.
Authors. 

Reviewer 2 Report

The manuscript addresses a topic of importance in engineering. It is of experimental type. The content and the results are well described, so that the methodology is easy to follow. My main concern is that the English language contains several errors. Most of them could have been easily corrected after a serious proofreading. In any case, the help of a professional editing service is highly recommended. Another concern is about the equations, which contains several quantities which have not been introduced. Besides, some equations are quite trivial (e.g. eq. 1) and could easily be removed.

Author Response

Dear Reviewer 2.

Thanks for your comments to the manuscript. It was performed changes including the professional language editing by MDPI in area of Computer science and Mathematics. To your comments was noted the biggest changes in the text in enclosed word. 
Thanks for our colaborating and your revision of the manuscript.
Yours sincerelly.
Authors.

Reviewer 3 Report

The proposed paper provides a inspection for tire defects. First of all, there are some typos, mostly regarding notation, exist.  Secondly, the term and phrases with English is very difficult to read. Editing improvement seems to be needed.

  1. In many cases, CNN can classify image regardless of orientation. The authors try to unfold the sidewall of tire. Have you tried to use image with unfold and compare them?
  2. Is there any affect by deformation of image to unfold and any problem with that?
  3. If the radius is known for each tire, is it easier to fix center and to estimate blue and red line?
  4. How the sync between vision data and laser data can be done? Does the size of image is adjusted without any centering between them?
  5. Generally, CNN is used for classification, which means that all kinds of abnormal type as well as normal should be trained. Hence, it is fatal to detect new type of abnormality.
  6. Considering the size of image, the number of used data seems to be needed more. 
  7. It seems better to show the comparison between only 2D or 3D and the merged data. 

Author Response

Dear Reviewer 3.

Thanks for your comments to the manuscript. It was performed changes including the professional language editing by MDPI in area of Computer science and Mathematics. To your comments was noted the biggest changes in the text in enclosed word. 
Thanks for our colaborating and your revision of the manuscript.
Yours sincerelly.
Authors.

Round 2

Reviewer 1 Report

As I said in previous reviews, the paper does not answer the questions in the title: unsupervised learning and deep learning The changes I see in the paper are only language.

Reviewer 3 Report

The requested modification and answers are covered by this version of paper.

This manuscript is a resubmission of an earlier submission. The following is a list of the peer review reports and author responses from that submission.

Round 1

Reviewer 1 Report

-The paper is a technical report in the field of sensing and imaging, with a research proposal and research ideas at a low scientific level. The paper is dealing with image, camera sensing concepts, image acquisition and processing and presents an application in industry, for surface sensing.

-The paper proposed some keywords for possible up to date methods as "unsupervised learning", "pattern recognition"  and "machine learning";  But, no details are provided related to the way these concepts are applied in the application.

So, it is strongly recommended to present in detail the way in which these concepts are used in the paper, so as to increase the scientific value of the paper.

-The application approached in the paper could be included in the field of defect detection and diagnosis, but the paper does not use basic concepts of this field, such as defect symptoms, residues, and so on.

Therefore, it is recommended to use concepts from the theory of defect detection and diagnosis to frame the work in the keyword "defect detection".

-The paper cites many articles, but which do not fall within the keywords of the paper. Thus, there is no basis for a concrete comparison of achievements in the field with the application described in the paper.

It is recommended to make a list of the improvements that the paper brings compared to the methods presented in each of the cited works. Eliminate the cited works that you do not use as a concrete comparison.

-Figures 1 ... 5 show some images taken experimentally. Image quality is low.

It is recommended to present some images with a better quality, which would allow the notification of the details discussed.

-The theoretical presentation of figures 1 ... 5 contains sentences that are in addition.

It is recommended to systematize this presentation to eliminate extra expressions.

-It is recommended to re-edit the equations.

-It is recommended to systematize the presentation of figures 7-11 in relation to the given equations.

- Section 2 presents the results of some experimental measurements. The paper does not describe the scientific methods applied to these measurements, it does not present the way of data processing. The part that responds to the keywords in the paper mentioned above is missing here. The paper shows in Fig. 18 a diagram of the inspection system, but only that. The paper gives some technical data of a numerical acquisition system, which are not important from the scientific perspective of a presentation.

-The part of results, discussions and conclusions does not have a basis for approaching the concepts stated by keywords if the paper does not show how they were used in solving the application.

Author Response

Dear reviewer. 
In advance, I would like to thank you for your inspiring comments. Based on this comments were performed the changes and added the paragraphs. Every change was highlighted by yellow to distinguish changes. 
I believe that my answers and the changes performed in the paper are satisfying for you.
In the case of unsatisfying answers, I would like make additional changes in the article.
I am looking forward to hearing from you.
Kind regards
Jaromír Klarák

Reviewer 2 Report

The manuscript is devoted to the solve the task of image recognizing.

The manuscript is well structured; it contains all sections for this type of publication. However, to my mind, the content of this paper should be revised since there are many mistakes and incorrectness.  Below, I present some of them.

  1. The abstract does not reflect the manuscript content. This is a short introduction. To my mind, it should be rewritten.
  2. English grammar is unsatisfactory. The article contains a lot of language mistakes and incorrectness.
  3. It will be better if before the Materials and Methods section the authors allocate unsolved parts of the general problem and add the section Problem Statement where they form clearly the solving task.
  4. The section Materials and Methods to my best mind should be divided into theoretical and practical parts. So, I think that it is necessary to add the Section Simulation.
  5. The quality of the equation representation is unsatisfactory too. They should be reformatted.
  6. The format of the figures also does not correspond to this type of publication.
  7. I think that figure 18 is the main authors’ achievements. This block chart should be described in detail.

Thus, to my best mind, the manuscript should be fully reformatted.

Author Response

(The authors gave the same response as above.)

Round 2

Reviewer 1 Report

The changes made by the authors do not significantly improve the work.

Reviewer 2 Report

Thanks for your correction and response. The manuscript really looks better. I have not any additional questions and remarks. I think that this manuscript can be published in the present form.